# Evaluation of pharmaceutically compounded oral caffeine on the impact of medication adherence and risk of readmission among preterm neonates: A single-center quasi-experimental study

Gul Ambreen[1]*, Manoj Kumar[2], Amin Ali[3], Syed Akbar Ali Shah[4], Syed Muzafar Saleem[2], Ayesha Tahir[2], Muhammad Sohail Salat[2], Muhammad Shahzad Aslam[5]*, Kashif Hussain[1]

1 Department of Pharmacy, Aga Khan University Hospital, Karachi, Pakistan, 2 Department of Paediatrics & Child Health, Aga Khan University, Karachi, Pakistan, 3 Department of Neonatology & Paediatrics, Dow University of Health Sciences, Karachi, Pakistan, 4 Department of Neonatology, Dr. Ruth K. M. Pfau, Civil Hospital Karachi, Karachi, Pakistan, 5 School of Traditional Chinese Medicine, Xiamen University Malaysia, Sepang, Malaysia

* gul.ambreen@aku.edu (GA); muhammad.salat@aku.edu (MSA)

**Data Availability Statement:** Data are available upon request due to ethical considerations. The

## Abstract

### Background

Caffeine is available in an ampoule, used via parenteral and enteral routes in preterm neonates to treat apnea of prematurity (AOP) in neonates of gestational age $\geq$ 35–40 weeks. A longer duration of therapy has a higher risk of medication non-adherence due to higher costs and inappropriate dosage forms. Pharmaceutically compounded oral caffeine (PCC) could be an appropriate alternate dosage form. The researchers aimed to determine the impact of PCC on medication-related factors influencing medication adherence (MA) and the frequency of hospital readmission with apnea (HRA) in preterm neonates.

### Methods

We conducted a single-center quasi-experimental study for this quality improvement project using PCC among the preterm neonates admitted in a tertiary care level-III NICU at the Aga Khan University Hospital Karachi, Pakistan, received caffeine therapy, and survived at discharge. The researchers compared pre-PCC data (April-December 2017) with post-PCC data (April-Dec 2018) each for nine months, with three months intervals (January-March 2018) of PCC formulation and implementation phase. The study was conducted according to the SQUIRE2.0 guidelines. The Data were collated on factors influencing MA, including the cost of therapy, medication refill rates, and parental complaints as primary outcome measures. The Risk factors of HRA were included as secondary outcomes.

data contain potentially identifying information.
Requests for data can be sent to the corresponding
author or to the institutional ethical committee at
erc.pakistan@aku.edu.

**Funding:** The author(s) received no specific
funding for this work.

**Competing interests:** Muhammad Shahzad Aslam
is the academic editor in PLOS ONE All other
authors declare that they have no conflict of
interest. This does not alter our adherence to PLOS
ONE policies on sharing data and materials.

## Results

After PCC implementation cost of therapy was reduced significantly from Rs. 97000.0 (729.0 USD) to Rs. 24500.0 (185.0 USD) (p<0.001), significantly higher (p<0.001) number of patients completed remaining refills (77.6% pre-phase vs 97.5% post-phase). The number of parental complaints about cost, ampoule usage, medication drawing issue, wastage, inappropriate dosage form, and longer duration of therapy reduced significantly in post-phase. HRA reduced from 25% to 6.6% (p<0.001). Post-implementation of PCC (RR 0.14; 95% CI: 0.07–0.27) was a significant independent risk factor for reducing HRA using a multivariate analysis model. Longer duration of caffeine therapy after discharge (RR 1.05; 95% CI: 1.04–1.04), those who were born in multiple births (RR 1.15; 95% CI: 1.15–1.15), and those who had higher number of siblings were other significant independent risk factors for HRA.

## Conclusions

PCC dispensation in the appropriate dosage form at discharge effectively reduced cost, non-adherence to therapy, and risk of hospital readmissions. This neonatal clinical and compounding pharmacist-led model can be replicated in other resource-limiting setting.

## Background

Apnea of prematurity (AOP) is a developmental disorder in preterm neonates due to immature respiratory control mechanisms [1,2]. It occurs in almost all neonates born at gestational age (GA)<29 weeks or birth weight (BW) <1000 g [3], in about 50% of the neonates with GA 30 to 32 weeks, and 7% neonates with GA 34–35 weeks [4]. AOP is associated with intermittent hypoxemia, therefore reported to affect the neurodevelopmental consequences and increased risk of retinopathy of prematurity (ROP) [5,6]. Furthermore, poor respiratory drive and AOP might be linked with prolonged mechanical ventilation duration and increased likelihood of extubation failure in neonates with respiratory distress [7].

Among all the methylxanthines, caffeine has a longer half-life, higher therapeutic index, and better enteral bioavailability; therefore, it is now established as a standard for treating AOP [8–10]. Its use is associated with shorter mechanical ventilation dependency and higher chances of extubation success in preterm neonates [8]. In most neonates, apneic episodes cease by term gestation [11], though apnea might persist beyond the term in infants born < 28 weeks' gestation [12]. Therefore, caffeine is used for preterm neonates for a longer duration in doses of 5–10 mg/kg/day once daily [4,6,8].

Due to the limited resources, stable infants are discharged from the hospital at comparatively lower body weight and GA. A study from the same center reported that infants were discharged from the hospital at BW 1286.4 (±219.6) g and corrected GA 33.0 (±3.3) weeks [13]. Adherence to caffeine therapy is highly required for the continuity of care and prevention of rehospitalization with apnea. Medication adherence (MA) [14] is defined as "The extent of a person's behavior taking medication, following a diet, and executing lifestyle changes, corresponds with agreed recommendations from a healthcare provider". Medication adherence focuses on patient compliance with drug regimen to achievement of desired therapeutic outcomes [15,16]. Acceptance, persistence, and execution are essential steps for achieving MA [17]. Therapy-related factors (e.g. safety and efficacy of therapy, length and complexity of treatment), and socioeconomic factors (e.g. financial difficulties) are strong determinants of MA [18,19].

## Challenges in the usage of caffeine- our local experience

Caffeine is being used in Pakistan since April 2017. Caffeine is commercially available in 20mg/ml ampoule used for intravenous and oral routes. A protocol was developed for caffeine use in our institute, which was applicable to all neonates born with GA <30 weeks and/or BW <1500 g. The duration of therapy is at least till the corrected GA of 35–40 weeks. Initially, a loading dose of 20 mg/kg/dose is given intravenously, followed by a 5-10/kg/day maintenance dose. Once the enteral feed is developed, caffeine citrate ampoule is administered through the oral route, which becomes a big challenge for the attendants at home. The chances of failure of MA at all steps (acceptance, persistence, and execution) were very high after discharge from the hospital due to the unavailability of caffeine in the appropriate dosage form. Most importantly, failure in execution, as this step involves accurate patient/attendant performance of medication use according to the recommendations including right dose [20,21]. The chance of over and under medication was very high, which is the primary concern with caffeine therapy in neonates due to the high potential of adverse effects and failure to achieve desired results, respectively [8,22]

Drug wastage and the high cost of the ampoule are other challenges and contributing factors to non-adherence to therapy and readmissions. The high cost of therapy per day (about Rs.2000 = 15USD/day) for the median 48 days (range 35-63days) is a significant economic burden on the family and possibly third parties that pay such as corporate groups and local health insurance companies. One ampoule contains caffeine citrate 20mg/ml; however, the amount used for each daily dose is a median of 7mg (ranges 5 to10 mg), and the rest of the content is discarded after a single use. Drawing the small volume of medication from the ampoule by parents/attendants at home also results in ampoule breakage and wastage. Finally resulted in reduced MA, patient/attendant dissatisfaction, and higher risk of HRA.

Pharmaceutical compounding is defined as a professional practice of a licensed pharmacist in which medications are prepared tailored to the individual patient's needs on the medication order by the licensed practitioner [23]. Over the past years, the role of traditional compounding practices has reduced with the accessibility of commercially manufactured drugs, but the impact and need for specialized compounding practices are increased [24]. The unavailability of a suitable dosage form is one of the significant reasons for compounding the medications for the pediatric and neonatal population [25]. A pediatric clinical pharmacist trained in pharmaceutical compounding more strongly believes that compounding is the solution to most of the challenges related to the unavailability of the appropriate commercial dosage form in this population. Moreover, this approach offers multiple benefits in patient-centered care and patient well-being with improved MA with the added value of cost-saving [26–30]. Therefore, the purpose of this study was to assess the association between the pharmaceutically compounded oral caffeine (PCC) use in preterm neonates and the therapy-related factors (e.g. safety and efficacy of therapy, length, and complexity of treatment), and socioeconomic factors (e.g. financial difficulties) influencing the MA. Moreover, the study aimed to determine patient, caffeine therapy, family, and intervention-related risk factors for hospital readmission with apnea among neonates.

## Methods

### Study design, settings, and duration

The impact of PCC implementation on predefined outcome measures was evaluated in the NICU of Aga khan university hospital (AKUH), a tertiary care setting in Karachi, Pakistan, with the facility of 24 bedded multispecialty tertiary care NICU, where about 1200 neonates

are admitted annually with the influx of very preterm high-risk newborns from all over the country. The AKUH pharmacy department has the facility of compounding services, where the pharmaceutical compounding practice is performed under the supervision of qualified pharmacists following the standard processes with nominal service charges in the outpatient pharmacy.

The researchers conducted a single-center quasi-experimental study for this quality improvement (QI) project and compared nine months (April-December 2017) of pre-PCC data with nine months (April-Dec 2018) of post-PCC data [31], with three months (January-March 2018) as the PCC formulation phase. The study was conducted according to the SQUIRE2.0 guidelines [32]. All components of PCC were progressively put in place through the training and feedback of all the stakeholders including prescribing physicians, order processing pharmacist, compounding pharmacist, medication administering nurses, and parents. Physician and nursing staff were also trained for proper communication with NICU pharmacists about caffeine therapy-related complaints.

## Sample size and population

The sample size was calculated using PASS version 11 by considering the hospital readmission as an outcome of MA. Literature suggests 20% of patients need rehospitalization due to low and intermediate MA [33]. We assumed a 60% reduction in readmission with apnea among infants due to PCC intervention. A total of 262 (131 newborns per group) were needed to detect this difference with at 80% power and 95% level of significance. All the preterm neonates born at AKUH admitted to NICU during the study period and prescribed caffeine for AOP were included in the study and were tracked until transfer to the step-down unit or discharge and rehospitalization. We excluded those neonates who were readmitted after the corrected GA of 35–40 weeks or with any reason other than apnea.

## Outcome measures and data collection

Data were retrieved retrospectively, for neonatal demographic and clinical characteristics, morbidity patterns, perinatal history, maternal steroidal exposure, duration of therapy, number of siblings, survival to discharge. We compared the efficacy and safety of compounded oral caffeine with the commercially available dosage form through clinical outcomes, including duration of mechanical ventilation (MV) and oxygen supplementation, postnatal age of first successful extubation, the incidence of bronchopulmonary dysplasia (BPD), necrotizing enterocolitis (NEC) and spontaneous intestinal perforation through the course of oral therapy. As a key measure of MA improvement, we selected therapy and socioeconomic-related factors, including caffeine therapy's cost and length of hospital stays. After discharge, the frequency of HRA (till the corrected GA of 35–40 weeks) was considered as the clinical outcome parameter. The medication refill data was checked from the pharmacy record to measure the adherence to therapy in both phases. The number of patient's attendant complaints concerning caffeine ampoule at home for oral administration was gathered to know the impact of change.

## Operational definitions

Successful extubation is defined as a postnatal age at which the infant was first extubated and remained extubated for at least more than 24 hrs. Ventilatory requirement/a day on oxygen supplementation is defined as at least 12 hrs need of MV/oxygen in 24 hrs. Bronchopulmonary dysplasia (BPD) and severe BPD are defined as the need for oxygen supplementation only and a fraction of inspired oxygen (FiO2) of $\geq 0.30$ and/or positive airway pressure for $\geq 28$ days and at 36 weeks PA, respectively. NEC is defined as Bell's stages II or III [34].

## Implementation of intervention

Standard Operating Procedure (SOP) was designed for pharmaceutical compounding of caffeine citrate 10 mg/ml oral solution and approved and adopted by the pharmacy compounding section AKUH in January 2018. Pharmacists, nurses, and physicians were given in service after making required changes in the Computerized Physician Order Entry (CPOE) system about the concentration and reduced cost. Implementation of practice change started in February 2018.

Due to the non-availability of caffeine anhydrous powder is in Pakistan, commercially available caffeine citrate ampoule was used for preparing oral compounded solution of caffeine citrate (10mg/mL) following standard procedures [35] and compounding references [36,37]. Since we used the commercially available finished products for PCC compounding, we only performed PCC's physical and microbiological testing. The pH of the commercially available ampoule was 5.1, and the pH of PCC was in the range of 4.9–5.2 over the 60 days study period stored at room temperature. No visible degradation and no change in the PCC solution's colour and odour were observed throughout the study period. A 30-day and 60-day microbial testing of PCC aqueous solution for oral use confirmed that compounded product was within the recommended acceptable criteria. To improve convenience and avoid wastage PCC was dispensed in an amber colour unbreakable plastic bottle in a final concentration of 10 mg/ml. This formulation is stable for two months at room temperature [36,37].

We measured medical adherence through the multi-measure approach, including reviewing caffeine prescription refill records and the self-reporting method [38]. The review of prescription refill records through the outpatient pharmacy computerized patient record system allowed us to calculate the number of caffeine ampoules or total volume of PCC solution dispensed against the total suspected duration of caffeine therapy after discharge. We collected the feedback through a self-reporting subjective technique, based on the parents' verbally communicated concerns and complaints to physicians, nurses, and pharmacists. During the implementation phase, all the healthcare team involved in this process was trained for this communication. These complaints were recorded at the time of discharge, at the clinic's revisit, and the outpatient pharmacy while caffeine refills. Finally, all these concerns and complaints were reported to the NICU pharmacist.

## Statistical analysis

All Categorical variables were presented as numbers and percentages. Continuous variables were expressed as Mean ± SD and Median (IQR), as appropriate. Chi-square test and independent-sample t-test were applied to compare pre-post implementation period for categorical and continuous variables, respectively. The generalized linear model with log link function was used to determine potential risk factors of hospital readmission with apnea. The bivariate analysis was conducted to determine the independent effect of each predictor on outcome Variables were accounted in multivariate analysis for adjustment where $p < 0.20$ in bivariate analysis and retain with $p<0.05$ using backward elimination. Results were reported as risk ratio (RR) and 95% confidence interval. AIC and BIC were used to assess the fit of the model. All data were analyzed through STATA 17.

**Ethics approval.**   This study was approved by the Ethical Review Committee of the AKUH (reference # 2019-2111-5600). The need for informed consent was waived due to the retrospective design of this study. All data collected was kept confidential.

# Results

## Demographic and clinical characteristics

A total of 1857 neonates were admitted to NICU, including 690 and 626 preterm neonates with GA of $\leq$ 37 weeks during the pre and post-implementation study phases respectively. A

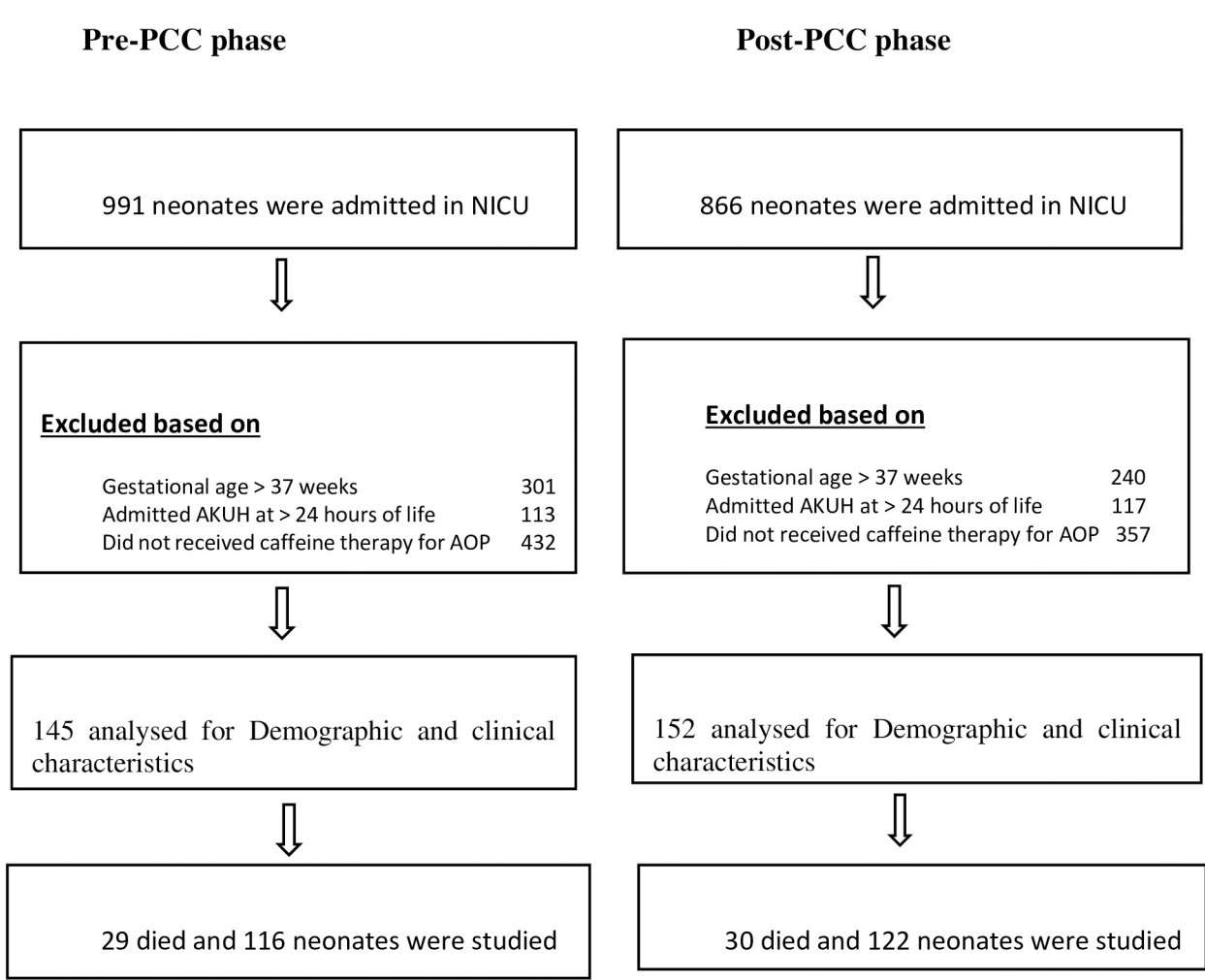

**Fig 1. Patient inclusion scheme.**

total of 297 were present in pre and post-implementation phases, who received loading and then maintenance caffeine as per our NICU caffeine protocol. In pre-phase 29 (20%) and post-phase 30 (19.7%) infants died, so finally, 116 and 122 neonates composed the pre-and post-implementation study groups, respectively (Fig 1). Neonatal demographic and clinical characteristics were comparable in both groups. However, the proportion of neonates on an MV, BW $\leq$1000 g, and GA of $\leq$27 weeks was higher in the post-implementation phase. Neonates in post-implementation phase were intubated at significantly smaller age than the neonates of pre-imp phase [5.2 ($\pm$ 3.8) hours vs 7.7 ($\pm$ 3.5) hours]. During both phases, 17 neonates of GA >31 to $\leq$ 36 weeks also received caffeine (Table 1).

## Safety and efficacy of caffeine in different dosage forms

Safety and efficacy-related variables of both the dosage forms are compared, and there is no significant difference in the postnatal age of first successful extubation, total days of oxygen, and MV need. BPD, NEC, and spontaneous perforation were comparable in both groups. The length of hospital stay was shorter in post-phase. After discharge, caffeine therapy was administered for 5–9 weeks in both groups. After PCC implementation cost of therapy was reduced significantly (p <0.001) from Rs. 97000.0 (729.0 USD) to Rs. 24500.0 (185.0 USD and hospital

**Table 1. Demographic and clinical characteristics of preterm neonates who received caffeine therapy in the pre and post-implementation phases.**

| Patient characteristics | Pre-PCC n = 145 (%) | Post-PCC n = 152 (%) | *p*-value |
|---|---|---|---|
| **Gender** | | | 0.79 |
| Male | 75(51.7) | 81(53.3) | |
| Female | 70 (48.3) | 71(46.7) | |
| Birth weight (g) @ | 1152.3 (321.5) | 1157.2 (359.2) | 0.90 |
| Birth weight (g) | | | 0.55 |
| ≤1000 | 59(40.7) | 65(42.8) | |
| 1001–1500 | 82(56.6) | 79(52.0) | |
| 1501–2500 | 4(2.8) | 6(3.9) | |
| ≥2501 | 0(0.0) | 2(1.3) | |
| GA (weeks) @ | 29.3 (1.8) | 29.1(2.0) | 0.50 |
| GA (weeks) | | | 0.16 |
| ≤27 | 14 (9.7) | 25 (16.4) | |
| >27 to ≤31 | 124 (85.5) | 117 (77.0) | |
| >31 to ≤36 | 7 (4.8) | 10 (6.6) | |
| Antenatal steroids (2 doses) | | | 0.62 |
| Yes | 69 (47.6) | 68 (44.7) | |
| No | 76 (52.4) | 84 (55.3) | |
| Emergency lower segment caesarean section | | | 0.13 |
| Yes | 57 (49.1) | 48(39.3) | |
| No | 59 (50.9) | 74 (60.7) | |
| 5-Minute Apgar score of <5 | | | 0.49 |
| Yes | 35 (24.1) | 42 (27.6) | |
| No | 110 (75.9) | 110 (72.4) | |
| Age at intubation (h) @ | 7.7 (3.5) | 5.2 (3.8) | < 0.001 |
| Received surfactant | | | 0.47 |
| Yes | 71 (49.0) | 68 (44.7) | |
| No | 74 (51.0) | 84 (55.3) | |
| Mechanical ventilation | | | 0.27 |
| Yes | 68 (46.9) | 81 (53.3) | |
| No | 77 (53.1) | 71 (46.7) | |
| **Outcome** | | | 0.92 |
| Step down | 110 (75.9) | 117 (76.9) | |
| Discharged home (from NICU) | 6 (4.1) | 5 (3.3) | |
| Died | 29 (20) | 30 (19.7) | |

Data is presented as n(%) unless otherwise indicated @ Data presented as mean SD

## Data presented as median(IQR); PCC = Pharmaceutically compounded oral caffeine; NICU = neonatal intensive care unit.

A p-value less than 0.05 (typically ≤ 0.05) is statistically significant.

readmissions with apnea reduced from 25% to 6.6% (p <0.001). The number of patients who completed all the remaining refills of caffeine doses increased significantly (77.6% pre-phase vs 97.5% post-phase) (p<0.05) (Table 2).

## Barriers to adherence with caffeine therapy

The dosage form-related barriers to adherence with caffeine therapy in pre and post-phases are evaluated in Table 3. The numbers and types of complaints related to caffeine regimen at discharge and during follow-up visits are compared for both phases (Figs 2 and 3).

At hospital discharge, parents shared complaints about the high cost of therapy was 96.6% vs 1.6%, parents refused to take discharge medication 2.6% vs 0.8% in the pre and post-implementation phase, respectively. At follow-up visits, 84.5% of parents had complaints of high

**Table 2. Safety and efficacy related clinical variables in pre and post implementation phases.**

| | Pre-PCC n = 116 | Post-PCC n = 122 | *p*-value |
|---|---|---|---|
| **Clinical variables before discharge** | | | |
| Postnatal age of first successful extubation (days)## | 8.0 (5.0–12.0) | 8.0 (6.0–19.0) | 0.094 |
| Total days on oxygen supplementation ## | 20.0 (12.5–27.0) | 17.5 (13.0–27.0) | 0.77 |
| Total days on Mechanical Ventilation ## | 19.0 (15.0–28.0) | 15.5 (9.0–25.0) | 0.011 |
| **Incidence of (n%)** | | | |
| ➤BPD | 13(11.2) | 16 (13.1) | 0.65 |
| ➤NEC | 15 (12.9) | 10 (8.2) | 0.23 |
| ➤Spontaneous perforation | 4 (3.4) | 2 (1.6) | 0.37 |
| **Length of NICU stay** | | | |
| mean SD | 16.6 ± 11.3 | 16.4 ± 10.4 | 0.84 |
| range (min, max) | (2–65) | (2–59) | |
| **Length of Hospital stay** | | | |
| mean SD | 32.2 ± 12.1 | 29.5 ± 11.6 | 0.073 |
| range (min, max) | (2–85) | (2–78) | |
| Discharge weight (g) @ | 1332.6 ± 306.0 | 1330.3 ± 410.0 | 0.96 |
| Corrected GA at discharge (weeks) @ | 31.4 ± 2.3 | 31.8 ± 2.4 | 0.24 |
| **Caffeine therapy-related variables** | | | |
| **Corrected GA for stopping caffeine therapy** | | | |
| mean SD | 38.1 ± 1.9 | 38.5 ± 2.0 | 0.12 |
| range (min, max) | (35.6–41.6) | (36.1–42.3) | |
| Post discharge duration of caffeine therapy (days) ## | 48.5 (35.0–57.0) | 49.0 (32.0–63.0) | 0.86 |
| Cost (PKR) for complete course of therapy ## | 97000.0 (70000.0–114000.0) | 24500.0 (16000.0–31500.0) | <0.001 |
| Cost (USD) for complete course of therapy ## | 729.0 (526.0–857.0) | 185 (120.0–237.0) | <0.001 |
| **Caffeine Refills dispensed for remaining duration after discharge** | | | <0.001 |
| Yes | 90 (77.6) | 119 (97.5) | |
| No | 26 (22.4) | 3 (2.5) | |
| **Clinical variables after discharge** | | | |
| **Hospital Readmissions with apnea (n%)** | | | <0.001 |
| Yes | 29 (25.0) | 8 (6.6) | |
| No | 87 (75.0) | 114 (93.4) | |

## Data presented median (IQR)

@ Data presented as mean SD; PKR = Pakistani rupee; PCC = Pharmaceutically compounded oral caffeine; BPD = bronchopulmonary dysplasia; NEC = necrotizing enterocolitis.

A p-value less than 0.05 (typically ≤ 0.05) is statistically significant.

cost, ampoule opening and medication with drawl issue, medication wastage, inappropriate dosage form, and longer duration of therapy. Only 15.5% of parents did not share any complaints. On the other hand, 78% of the post-implementation phase had no complaint.

## Regression analysis

Neonates in the post-PCC group were at lower risk for hospital readmission with apnea (HRA) (p <0.0001; RR 0.26; 95% CI: 0.13–0.55) compared to the pre-intervention group with the bivariate level of analysis. Neonates with a longer duration of therapy (p<0.0001; RR: 1.05; 95% CI: 1.03–1.06), multiple births (p<0.0002; RR: 2.50; 95% CI: 1.40–4.46), those who were discharged on oral medication (p<0.002; RR: 0.34; 95% CI: 0.17–0.68), those who were

**Table 3. Impact of PCC intervention on dosage form related barriers to adherence with caffeine therapy.**

| Factors | Pre-PCC | Post-PCC |
| --- | --- | --- |
| Cost (PKR) of therapy (per day) | 2000 (19.4$) | 494 (4.8$) |
| Dosage form used for oral route | Ampoule | A syrup bottle |
| Accidental Wastage while dose administration | High (Breakable ampoule) | Very low (Unbreakable bottle) |
| Medication wastage | 75% (5mg used and 15mg wasted) | 0% (only required about drawn) |
| Dosage volume | Smaller | Larger |

PKR = Pakistani rupee; PCC = Pharmaceutically compounded oral caffeine.

discharged on >4 oral medication (p<0.298; RR: 1.43; 95% CI: 0.73–2.78), those delivered through emergency lower segment cesarean section (p<0.0001; RR: 6.51; 95% CI: 2.38–17.80), > 5 visits for medication refills (p<0.035; RR: 2.30; 95% CI: 1.06–4.97) and those who were not dispensed complete refills (p<0.0001; RR: 3.46; 95% CI: 1.96–6.11) were more likely to HRA than the comparison group. The risk of HRA was also higher in female participants and those who had higher number of siblings but was not statistically significant. Weight at discharge showed no significant association with HRA at the bivariate level.

Post-implementation of PCC (RR 0.14; 95% CI: 0.07–0.27) was a significant independent risk factor for reducing HRA using a multivariate analysis model controlling confounders. Longer duration of caffeine therapy after discharge (RR 1.05; 95% CI: 1.04–1.04), those who were born in multiple births (RR 1.15; 95% CI: 1.15–1.15), and those who had higher number of siblings were also retained their significance as independent risk factors for HRA (Table 4).

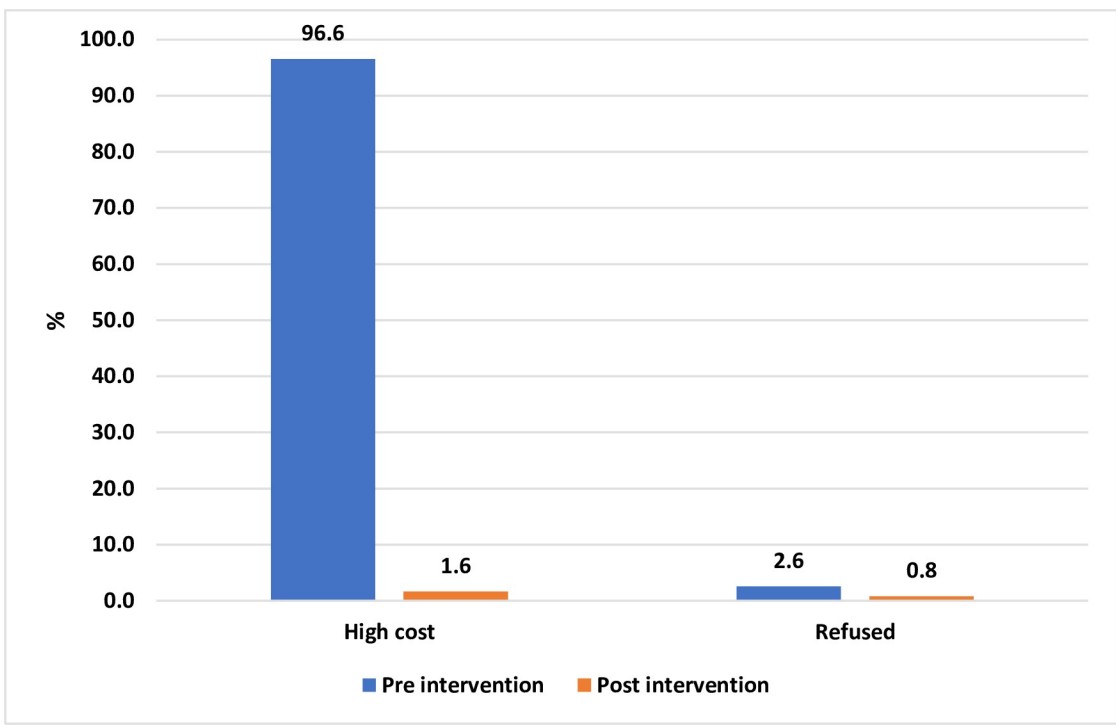

**Fig 2. Caffeine therapy related complaints at discharge.**

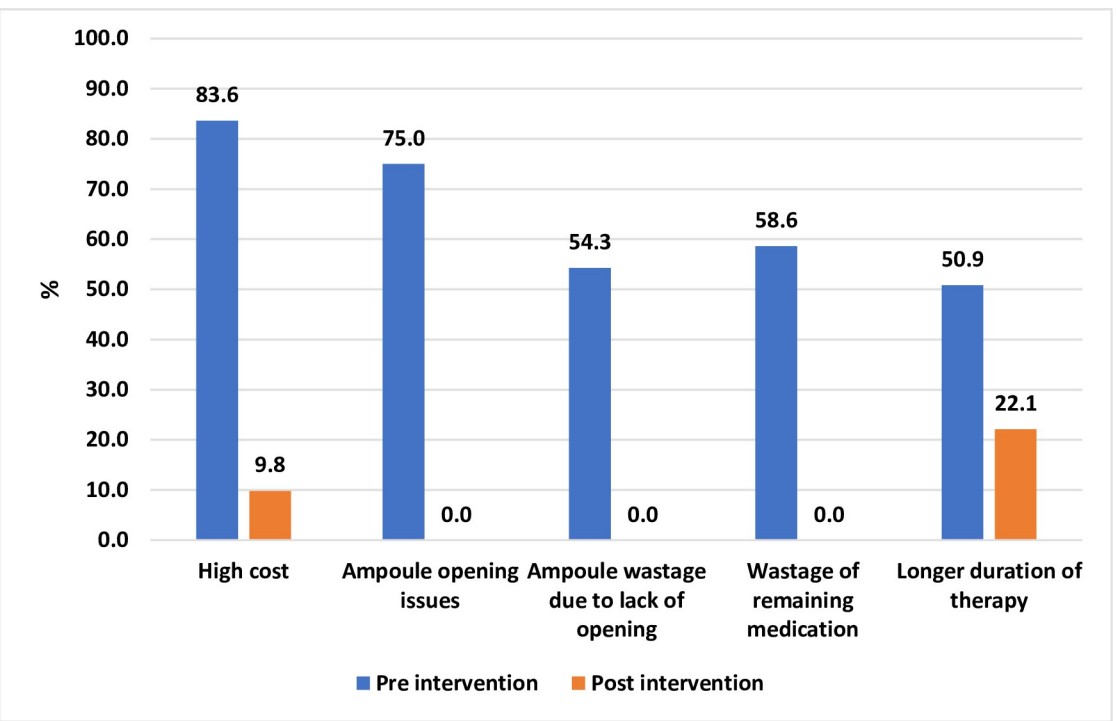

**Fig 3. Caffeine therapy related complaints at follow up visits and pharmacy visit.**

## Discussion

With the availability of oral caffeine in the appropriate dosage form, this QI initiative proved efficacious resulting in reduced cost of caffeine therapy, dosage form-related complaints, and improved caffeine therapy adherence that finally reduced the number of readmissions with apneic spells in our neonatal care setting. To the best of our knowledge, this is the first study addressing the impact of cost reduction and appropriate dosage form of caffeine on therapy adherence after hospital discharge and hospital readmission with apnea. Our results are like the previous literature showing the reduced readmission rates with improved MA [19,33].

After the availability of mass-manufactured drugs, the role of compounding pharmacy is now more specialized to compound the medications for an individual patient or a group of patients with the same need, as preterm neonates in the case of our study. This specialty of the pharmacist, with in-depth knowledge of compounding, enhanced the value of pharmacists in a clinical setting and improved the pharmacist-patient communication [39,40]. Pharmaceutical compounding of medicines offers benefits in terms of developing a patient-centered health-care system and ultimately human well-being [26]. Furthermore, cost savings and improved MA values are strongly associated with compounded medications [26].

Previous literature also suggests that the same strategy of compounding the patient-specific medicines in a pharmacy setting has been adopted to overcome the unavailability of commercial medicinal products in a suitable strength for pediatric patients [26,33,41]. The use of standard compounding reference [36,37] and techniques for compounding oral caffeine [35] proved to be an effective and safe practice. Although in the post-PCC phase there were a higher number of neonates who had BW ≤1000 g and who needed MV. However, PCC was found equally effective as the neonates who received PCC had an insignificant difference or improved results in terms of total days on oxygen and ventilatory requirement and the postnatal age of

**Table 4. Contributing factors for hospital readmission with apnea in both phases.**

| | Hospital readmission with apnea | | Bivariate | | | Multivariate | | | |
| | Yes | No | | | | | | | |
| | N = 37 | N = 201 | RR | CI | P value | RR | CI | P value | |
| **Group** | | | | | | | | | |
| Pre-PCC intervention | 29 (78.4%) | 87 (43.3%) | Ref. | | | | Ref. | | |
| Post-PCC intervention | 8 (21.6%) | 114 (56.7%) | 0.26 | 0.13 | 0.55 | <0.0001 | 0.14 | 0.07 | 0.27 | <0.0001 |
| **Gender** | | | | | | | | | |
| Male | 19 (51.4%) | 119 (59.2%) | Ref. | | | | | | |
| Female | 18 (48.6%) | 82 (40.8%) | 1.31 | 0.72 | 2.36 | 0.374 | | | |
| **Duration of caffeine therapy after discharge** | | | | | | | | | |
| Days (mean ± SD) | 64.9 ± 12.8 | 43.8 ± 18.7 | 1.05 | 1.03 | 1.06 | <0.0001 | 1.04 | 1.04 | 1.04 | <0.0001 |
| **Weight at discharge** | | | | | | | | | |
| (g) Median (IQR) | 1400.0 (1050.0–1610.0) | 1510.0 (1130.0–1620.0) | 0.999 | 0.998 | 1.000 | 0.210 | | | |
| **Multiple Births** | | | | | | | | | |
| yes | 15 (40.5%) | 36 (17.9%) | 2.50 | 1.40 | 4.46 | 0.002 | 1.15 | 1.15 | 1.15 | < 0.0001 |
| no | 22 (59.5%) | 165 (82.1%) | Ref. | | | | Ref. | | |
| **Poly-medication at discharge to home (number of medications)** | | | | | | | | | |
| ≤ 2 (oral) | 15 (40.5%) | 47 (23.4%) | Ref. | | | | | | |
| 3–4 (oral) | 12 (32.4%) | 135 (67.2%) | 0.34 | 0.17 | 0.68 | 0.002 | | | |
| >4 (oral) | 10 (27.0%) | 19 (9.5%) | 1.43 | 0.73 | 2.78 | 0.298 | | | |
| **Number of siblings** | | | | | | | | | |
| PG | 21 (56.8%) | 83 (41.3%) | Ref. | | | | Ref. | | |
| 1–2 | 9 (24.3%) | 68 (33.8%) | 0.58 | 0.28 | 1.19 | 0.138 | 0.72 | 0.72 | 0.72 | < 0.0001 |
| > = 3 | 7 (18.9%) | 50 (24.9%) | 0.61 | 0.28 | 1.34 | 0.219 | 0.75 | 0.75 | 0.75 | < 0.0001 |
| **Emergency lower segment caesarean section** | | | | | | | | | |
| yes | 4 (10.8%) | 101 (50.2%) | Ref. | | | | | | |
| no | 33 (89.2%) | 100 (49.8%) | 6.51 | 2.38 | 17.80 | <0.0001 | | | |
| **Number of visits to outpatient pharmacy for caffeine refill per each patient** | | | | | | | | | |
| <4 | 9 (24.3%) | 81 (40.3%) | Ref. | | | | | | |
| 4–5 | 14 (37.8%) | 73 (36.3%) | 1.61 | 0.73 | 3.52 | 0.234 | | | |
| >5 | 14 (37.8%) | 47 (23.4%) | 2.30 | 1.06 | 4.97 | 0.035 | | | |
| **Caffeine Refills dispensed for remaining duration after discharge (Y/N)** | | | | | | | | | |
| yes | 25 (67.6%) | 184 (91.5%) | Ref. | | | | | | |
| no | 12 (32.4%) | 17 (8.5%) | 3.46 | 1.96 | 6.11 | <0.0001 | | | |

RR = relative risk: CI = confidence intervals: PG = primary gravida; PCC = Pharmaceutically compounded oral caffeine.

A p-value less than 0.05 (typically ≤ 0.05) is statistically significant.

first successful extubation than the neonates of pre-PCC phase. In addition, our results also report the safety of PCC with the insignificant difference in the incidences of BPD, NEC, and spontaneous intestinal perforation in pre and post PCC implementation phases.

The availability of caffeine ampoules was very limited in a few hospital settings in Pakistan during the study period of both phases, so the AKUH outpatient pharmacy was the only option for the refill of caffeine prescriptions for all the AKUH discharged patients. Reviewing

prescription refill records is an indirect, simple, low-cost, and highly accurate method to assess multidrug adherence for various formulations. Furthermore, we achieved real-time feedback through a low-cost self-reporting subjective technique simply through the verbally communicated concerns and complaints [38]. Through this method, we could get individual patient/attendant concerns about therapy and potential factors of non-adherence to caffeine therapy after neonatal discharge from the hospital. Which subsequently helped for appropriate intervention to address the voice of the customer [42,43].

Worldwide, MA is significantly affected by the cost of therapy [44]. Despite the high number of complaints about the high cost of medicine and concerns about the longer duration of therapy, refill completion was found in majority of cases in the pre-PCC phase. That might be associated with the parents' natural response to fulfill the child's medicinal need [45]. However, despite getting refills, accuracy in execution was a significant barrier to MA. With the implementation of PCC, we could significantly reduce the cost of therapy. Overall, a significant reduction was observed in HRA after the PCC implementation. We can correlate the MA to better access to care with low-cost caffeine therapy that finally resulted in reduced HRA. We believe that HRA could be directly correlated with the suitable dosage form, i.e. oral solution in a bottle rather than an ampoule. Which also involves the special skills and technique to draw the exact volume from ampoule and results in failure in execution [20,46].

Pediatric patients (birth till 16 or 18 years) have different pharmacotherapy requirements than adults [47]. In high and low-middle-income countries (LMICs), the key pediatric age-appropriate dosage formulation parameters include correct, flexible dosing possibility with ease of administration and acceptable palatability [47]. In clinical settings, caffeine citrate is used for treating AOP in neonates therefore this study emphasizes the availability of the proposed dosage form to avoid wastage and minimize the cost to improve patient access. This approach may have comprehensively improved medicine's "intended use" in clinical and domiciliary practices [48].

In this project of PCC implementation, all the existing resources were used effectively and focused on the development of the compounded product in the most appropriate dosage form that parents/attendant's in-home environment can conveniently use that improved the MA and reduced the wastage. Our results are comparable with previous pharmaceutical compounding-based literature [19,26,33]. This QI project paves the way for more pediatric pharmaceutical care studies in low-income settings with financial challenges.

Another aspect of this cost-effective study was customer satisfaction, measured through customer complaints and feedback. Significantly reduced complaints concerning high cost and inappropriate dosage form of the caffeine had improved customer satisfaction. Among the neonates who were discharged on caffeine therapy, this QI approach significantly reduced the complaints in the post-PCC phase at the time of discharge and revisit to the hospital. The highest number of parents' complaints were about the cost and the inappropriate dosage form. Suboptimal adherence leads to poorer clinical outcomes and increased health care costs [49,50].

In our study MA to caffeine therapy was assessed through completion of refills, patients' complaints, and HRA; it offered the potential for targeting interventions to improve adherence. Consistent with previous literature from low source settings [51,52], the results of this study showed the higher risk of HRA due to non-adherence with caffeine therapy, with multiple births, infants discharged on poly-medication, more prolonged therapy, and completion of refill. After the adjustment of confounders in the multivariate model and the PCC intervention, only polypharmacy and low refill rate persisted as significant risk factors for HRA.

Despite all the achievements, some limitations are also linked with this study. It is a single-centered study with limited duration and generalizability. The quasi-experimental design is

associated with inherent limitations, including the potential for confounding bias. However, we did not find a significant difference in the patients' characteristics in the pre-PCC and post-PCC phases. Still, differences in unmeasured factors may exist between the groups. We could not evaluate the parents' financial status and educational background that could also affect medication non-adherence associated with HRA. For the self-reporting subjective technique, no validated tool was used, it was based on patients' complaints. Nonetheless, our study assessed caffeine therapy adherence with multiple methods, which strengthens the study [53]. This study could serve as an example to other LMICs where availability of oral caffeine and cost of therapy are barriers to MA and the researchers are interested in analyzing the potential effectiveness of their interventions.

## Conclusion

Medication adherence improved in infants discharged on caffeine therapy in a resource-limited setting with the intervention of pharmaceutically compounded oral caffeine solution with low cost and most appropriate dosage form, resulting in parents' convenience, minimum wastage, and access to care. It also helped to increase patient adherence to therapy by increasing refill rate, reducing complaints, and finally reducing hospital readmissions. This simple, evidence-based intervention with the strong involvement and empowerment of clinical and compounding pharmacists was executed without extra resources and, therefore, can serve as the classical model for access and continuity of medical care in developing countries.

## Supporting information

**S1 File.**
(DOC)

## Author Contributions

**Conceptualization:** Gul Ambreen, Kashif Hussain.

**Data curation:** Manoj Kumar, Amin Ali, Syed Akbar Ali Shah, Syed Muzafar Saleem, Ayesha Tahir.

**Formal analysis:** Muhammad Sohail Salat, Kashif Hussain.

**Investigation:** Gul Ambreen, Muhammad Sohail Salat.

**Methodology:** Gul Ambreen, Kashif Hussain.

**Project administration:** Gul Ambreen, Kashif Hussain.

**Resources:** Gul Ambreen, Muhammad Shahzad Aslam, Kashif Hussain.

**Software:** Kashif Hussain.

**Supervision:** Gul Ambreen, Amin Ali, Muhammad Sohail Salat.

**Validation:** Manoj Kumar, Muhammad Shahzad Aslam, Kashif Hussain.

**Visualization:** Muhammad Sohail Salat.

**Writing – original draft:** Gul Ambreen.

**Writing – review & editing:** Muhammad Sohail Salat, Muhammad Shahzad Aslam, Kashif Hussain.

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
