## [Decision Letter · Decision Letter 0]

22 Jun 2022

PONE-D-22-07283Evaluation of pharmaceutically compounded oral caffeine on the impact of medication adherence and risk of readmission among preterm neonates: A single-center quasi-experimental studyPLOS ONE

Dear Dr. Ambreen,

Thank you for submitting your manuscript to PLOS ONE. After careful consideration, we feel that it has merit but does not fully meet PLOS ONE’s publication criteria as it currently stands. Therefore, we invite you to submit a revised version of the manuscript that addresses the points raised during the review process.

We look forward to receiving your revised manuscript.

Kind regards,

Francesca Baratta, PharmD, PhD

Academic Editor

PLOS ONE

Journal Requirements:

“Muhammad Shahzad Aslam received this award. This work was supported by the Research Management Center, Xiamen University Malaysia, grant number XMUMRF/2020-C6/ITCM/0005.”

“Muhammad Shahzad Aslam is the academic editor in PLOS ONE .

All other authors declare that they have no conflict of interest.”

Reviewers' comments:

Reviewer's Responses to Questions

**Comments to the Author**

1. Is the manuscript technically sound, and do the data support the conclusions?

Reviewer #1: Yes

Reviewer #2: Yes

2. Has the statistical analysis been performed appropriately and rigorously? 

Reviewer #1: Yes

Reviewer #2: Yes

3. Have the authors made all data underlying the findings in their manuscript fully available?

Reviewer #1: Yes

Reviewer #2: Yes

4. Is the manuscript presented in an intelligible fashion and written in standard English?

Reviewer #1: Yes

Reviewer #2: Yes

5. Review Comments to the Author

Reviewer #1: GENERAL COMMENTS

The findings of the research are clearly presented and add to the body of knowledge. The submission should be accepted after minor corrections.

ABSTRACT

Well-written.

Preferably, arrange the keywords in alphabetical order.

BACKGROUND

Page 3 (line 24) and Page 4 (Line 1): The last sentence lacks fluidity.

METHODS

Page 7, Line 6: Data collection and …

Page 8, Line 10 – 12: Sentence is ambiguous. Rephrase.

RESULTS

Well-written.

DISCUSSION

Well-written.

CONCLUSION

Apt.

REFERENCES

The authors need to re-visit the References and ensure there is consistency with the Journal’s guidelines.

TABLES AND FIGURES

Tables that have p-values should include the level of significance as a footnote.

Reviewer #2: Overall well written report. Kudos to authors for a good job done. Minor changes suggested

Page 1

Line 6: suggest replace "until corrected" with "in neonates of"

Line 8-9: Suggest replace sentence with "Pharmaceutically compounded oral caffeine (PCC) could be an appropriate alternate dosage form"

Line 20: replace "added" with "included"

Line 24: pre-phase refills are higher than post-phase refill?

Page 2

Line 10: replace with "in other Resource-limited settings"

Page 7

Line 2: replace "was" with "were"

Line 6.: Replace Date with Data

Results

Give subheadings to separate the various sections for better clarity.

Page 10

line 18

Caffeine refills pre-PCC has been swapped with Post PCC making it seem refills for caffeine were higher pre-phase than post-phase which is not the case in table 2. Kindly reconcile.

Discussion

Page 11

Line 24: replace "results in" with " , resulting in"

page 12

Line 2: Kindly avoid making statements along the lines of being the first which may be refuted in future. May replace with "To the best of our knowledge, this is first study addressing........"

Line 19, "needed required" replace with either one of them

place comma (,) between MV and PCC

Suggest sentence be split into two or more to improve clarity. from Line 18 to 22

Page 13

line 5-6 Move sentence to limitations section as strengths and limitations of study

Line 16: Repeating results in discussion. Could use fraction or descriptive terms to describe 78% such as "majority...." etc. Similar case in Line 19

6. PLOS authors have the option to publish the peer review history of their article (what does this mean?). If published, this will include your full peer review and any attached files.

Reviewer #1: No

Reviewer #2: No

---

## [Author Response · Author response to Decision Letter 0]

5 Sep 2022

We are grateful to all the reviewers and editor for giving welcoming and motivating comments

regards 

Dr. Gul Ambreen

---

## [Editor Report · Decision Letter 1]

21 Sep 2022

Evaluation of pharmaceutically compounded oral caffeine on the impact of medication adherence and risk of readmission among preterm neonates: A single-center quasi-experimental study

PONE-D-22-07283R1

Dear Dr. Ambreen,

We’re pleased to inform you that your manuscript has been judged scientifically suitable for publication and will be formally accepted for publication once it meets all outstanding technical requirements.

Kind regards,

Francesca Baratta, PharmD, PhD

Academic Editor

PLOS ONE

---

## [Editor Report · Acceptance letter]

28 Oct 2022

PONE-D-22-07283R1 

*Evaluation of pharmaceutically compounded oral caffeine on the impact of medication adherence and risk of readmission among preterm neonates: A single-center quasi-experimental study*

Dear Dr. Ambreen:

I'm pleased to inform you that your manuscript has been deemed suitable for publication in PLOS ONE. Congratulations! Your manuscript is now with our production department. 

Kind regards, 

on behalf of

Dr. Francesca Baratta 

Academic Editor

PLOS ONE